# Outbreak of NDM-1-Producing *Escherichia coli* in a Coronavirus Disease 2019 Intensive Care Unit in a Mexican Tertiary Care Center

Oscar A. Fernández-García,[a] María F. González-Lara,[b] Marco Villanueva-Reza,[a] Nereyda de-León-Cividanes,[a] Luis F. Xancal-Salvador,[b] Verónica Esteban-Kenel,[b] Ayleen Cárdenas-Ochoa,[b] Axel Cervantes-Sánchez,[b] Areli Martínez-Gamboa,[b] Eric Ochoa-Hein,[c] Arturo Galindo-Fraga,[c] Miriam Bobadilla-del-Valle,[b] José Sifuentes-Osornio,[d] Alfredo Ponce-de-León[a]

[a]Infectious Diseases Department, Instituto Nacional de Ciencias Médicas y Nutrición Salvador Zubirán, Mexico City, Mexico
[b]Clinical Microbiology Laboratory, Instituto Nacional de Ciencias Médicas y Nutrición Salvador Zubirán, Mexico City, Mexico
[c]Hospital Epidemiology and Healthcare Quality Control, Instituto Nacional de Ciencias Médicas y Nutrición Salvador Zubirán, Mexico City, Mexico
[d]Department of Medicine, Instituto Nacional de Ciencias Médicas y Nutrición Salvador Zubirán, Mexico City, Mexico

**ABSTRACT** Emergency department areas were repurposed as intensive care units (ICUs) for patients with acute respiratory distress syndrome during the initial months of the coronavirus disease 2019 (COVID-19) pandemic. We describe an outbreak of New Delhi metallo-$\beta$-lactamase 1 (NDM-1)-producing *Escherichia coli* infections in critically ill COVID-19 patients admitted to one of the repurposed units. Seven patients developed infections (6 ventilator-associated pneumonia [VAP] and 1 urinary tract infection [UTI]) due to carbapenem-resistant *E. coli*, and only two survived. Five of the affected patients and four additional patients had rectal carriage of carbapenem-resistant *E. coli*. The *E. coli* strain from the affected patients corresponded to a single sequence type. Rectal screening identified isolates of two other sequence types bearing $bla_{NDM-1}$. Isolates of all three sequence types harbored an IncFII plasmid. The plasmid was confirmed to carry $bla_{NDM-1}$ through conjugation. An outbreak of clonal NDM-1-producing *E. coli* isolates and subsequent dissemination of NDM-1 through mobile elements to other *E. coli* strains occurred after hospital conversion during the severe acute respiratory syndrome coronavirus 2 (SARS-CoV-2) pandemic. This emphasizes the need for infection control practices in surge scenarios.

**IMPORTANCE** The SARS-CoV-2 pandemic has resulted in a surge of critically ill patients. Hospitals have had to adapt to the demand by repurposing areas as intensive care units. This has resulted in high workload and disruption of usual hospital workflows. Surge capacity guidelines and pandemic response plans do not contemplate how to limit collateral damage from issues like hospital-acquired infections. It is vital to ensure quality of care in surge scenarios.

**KEYWORDS** COVID-19, carbapenemase, *E. coli*, carbapenem-resistant enterobacteria, NDM-1, ventilator-associated pneumonia, hospital-acquired infection

Address correspondence to Alfredo Ponce-de-León, alf.poncedeleon@gmail.com.

The authors declare no conflict of interest.

The coronavirus disease 2019 (COVID-19) pandemic has created a surge of critically ill patients. Work burdens, patient isolation, the use of personal protective equipment (PPE), and disruption of usual practices have influenced the incidence of hospital-acquired infections (HAIs). A prospective cohort study involving 148 health care facilities in the United States reported 60% more central line-associated bloodstream infections, 43% more catheter-associated urinary tract infections (UTIs), and 44% more cases of methicillin-resistant *Staphylococcus aureus* bacteremia from March 2020 to December 2020 than would be expected based on predictions. There were also 24% more multidrug-resistant-organism infections. Higher COVID-19 discharge rates were

**TABLE 1** Antimicrobial susceptibility of the index *E. coli* isolate

| Antimicrobial | MIC ($\mu$g/mL) | Interpretation[a] |
|---|---|---|
| Cefoxitin | >64 | R |
| Ceftriaxone | >64 | R |
| Ampicillin-sulbactam | >32 | R |
| Piperacillin-tazobactam | >128 | R |
| Imipenem | 8 | R |
| Meropenem | >16 | R |
| Amikacin | 2 | S |
| Tigecycline | 0.5 | S |
| Colistin | 0.25 | I |

[a]R, resistant; I, intermediate; S, susceptible.

associated with higher rates of hospital-acquired and multidrug-resistant organism infections (1).

Carbapenem-resistant (CR) *Enterobacteriaceae* (CRE) can cause outbreaks in health care settings. Carbapenemase-producing CRE are associated with worse outcomes than non-carbapenemase-producing CRE (2). Carbapenemases are encoded by mobile genetic elements that can disseminate among bacteria (3–5). We describe an outbreak of New Delhi metallo-$\beta$-lactamase 1 (NDM-1)-producing *Escherichia coli* among COVID-19 patients receiving care in a reconverted intensive care unit (ICU).

The outbreak occurred in a tertiary care hospital in Mexico City. The study center was converted into a COVID-19-dedicated facility in March 2020. The hospital's ICU capacity was expanded from 14 to 42 beds by repurposing two additional areas of the emergency department with 8 and 20 additional beds, respectively. At the time of the outbreak, three ICUs were simultaneously caring for COVID-19 patients. The ICUs were not physically connected, and health care personnel were not shared between the units.

A CR *E. coli* strain was isolated from an endotracheal aspirate (ETA) culture taken from a 47-year-old man with ventilator-associated pneumonia (VAP). The isolate's antibiogram is shown in Table 1. Phenotypic tests detected the presence of a metallo-$\beta$-lactamase (MBL) (6, 7). During the next 10 days, CR *E. coli* was isolated from an additional 6 patients in the same ICU. Following documentation of the outbreak, the infection control committee decided to screen all patients in COVID-19 ICUs for CR *E. coli* carriage (8). Screening was performed at one time point in the week following detection of the index case.

## RESULTS

Between 2 June and 12 June 2020, seven patients had infections due to CR *E. coli*. Six of them developed VAP and one had a complicated urinary tract infection (UTI). Table 2 describes the patients' characteristics and outcomes. Five patients were treated with high-dose tigecycline and colistin combination therapy. Meropenem was added in 2 of these 5 patients due to concomitant isolation of *Pseudomonas aeruginosa* and *Klebsiella variicola*. Daptomycin was used in 1 of these 5 patients for concomitant *Enterococcus faecium* bacteremia. Two patients received meropenem monotherapy; they died before CR *E. coli* had been identified.

Two cases survived (29%). Septic shock was the cause of death in the five deceased patients (71%). Death occurred at a median of 9 days (range, 1 to 19 days) after cultures were taken.

Metallo-$\beta$-lactamase production was phenotypically documented in all seven isolates. Five isolates were available for molecular testing. Four had $bla_{NDM-1}$ detected, and one had a negative real-time PCR result. These five isolates had multilocus sequence typing (MLST) performed; all corresponded to sequence type 361 (ST361).

Rectal swabs were taken from 34 patients in the three COVID-19 ICUs. Nine patients were found to be colonized by CR *E. coli*. All nine were admitted to the same ICU. This

**TABLE 2** Characteristics and outcomes of affected patients[a]

| Patient | Gender/age (yrs)[c] | Days on IMV[d] | Clinical sample[e] | mCIM/eCIM[f] | $bla_{NDM-1}$ real-time PCR[g] | MLST | Treatment[h] | Outcome |
|---|---|---|---|---|---|---|---|---|
| E*[b] | M/47 | 13 | ETA | +/+ | Positive | ST361 | MEM/DAP/TGC/CST | Discharge |
| D | F/55 | 23 | ETA | +/+ | Positive | ST361 | TGC/CST | Death |
| A | M/46 | 15 | ETA | +/+ | Negative | ST361 | TGC/CST | Death |
| B | M/39 | 22 | ETA | +/+ | Positive | ST361 | TGC/CST | Discharge |
| F | F/65 | 6 | ETA | +/+ | Positive | ST361 | MEM/TGC/CST | Death |
| I | F/38 | 5 | ETA | +/+ | ND[i] | ND | MEM | Death |
| C | M/55 | 13 | Urine | +/+ | ND | ND | MEM | Death |

[a]Characteristics and outcomes of affected patients. Letters represent individual patients and match those of Figure 1.
[b]E*, Index patient.
[c]M, male; F, female.
[d]IMV, invasive mechanical ventilation.
[e]ETA, endotracheal aspirate.
[f]mCIM, modified carbapenemase inactivation method; eCIM, EDTA-supplemented carbapenemase inactivation method; +, positive.
[g]NDM-1, New Delhi metallo-b-lactamase 1.
[h]MEM, meropenem; DAP, daptomycin; TGC, tigecycline; CST, colistin.
[i]ND, not done.

amounts to a 26% (9/34) prevalence of colonization in all critically ill patients and 45% (9/20) in those admitted to the affected ICU.

Five of the nine colonized patients belonged to the group with clinically documented infections. Their rectal isolates harbored $bla_{NDM-1}$. Four additional patients were identified as CR *E. coli* carriers; two had $bla_{NDM-1}$ detected and two harbored $bla_{OXA-48-like}$.

The seven rectal isolates harboring $bla_{NDM-1}$ were analyzed by MLST. Five corresponded to ST361, one to ST405, and one to a sequence type not previously described.

The five endotracheal and seven rectal $bla_{NDM-1}$-bearing *E. coli* isolates that underwent MLST were subjected to pulsed-field gel electrophoresis (PFGE). The ST361 isolates clustered in two branches with >95% estimated similarity, while the other two $bla_{NDM-1}$-carrying isolates (ST405 and ST unknown) diverged earlier (Fig. 1).

Replicon typing identified IncFII in all three sequence types carrying $bla_{NDM-1}$ (ST361, ST405, and ST unknown). The plasmid had a size of 87 kb. Conjugation experiments managed to transfer the 87-kb plasmid to transconjugants. High conjugation frequencies ($2.3 \times 10^7$ to $1.33 \times 10^8$ transconjugants per donor cell) were achieved (Fig. 2). $bla_{NDM-1}$ carriage on transconjugants was confirmed with endpoint PCR.

## DISCUSSION

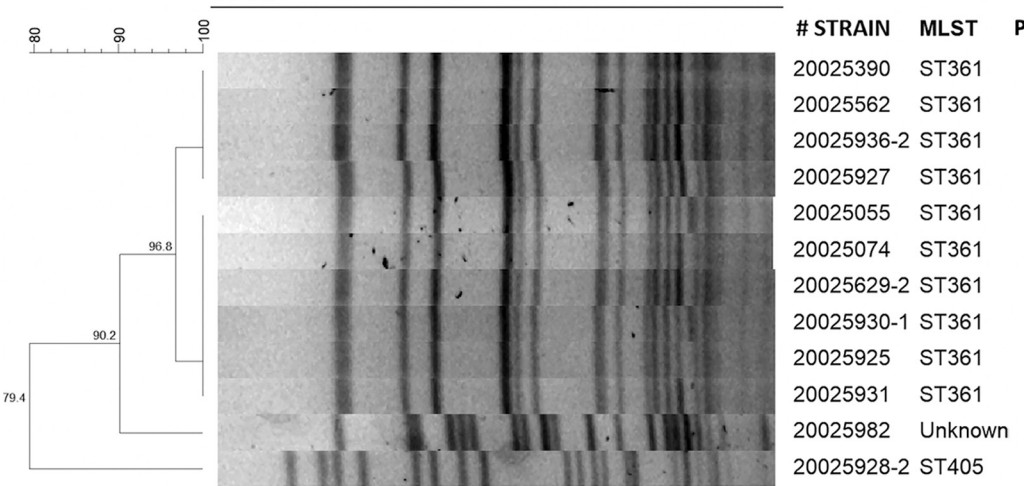

**FIG 1** Genetic relatedness of $bla_{NDM-1}$ carrying *E. coli* isolated and sequence types. Letters represent patients. Repeated letters mean samples came from the same patient. E*, index patient; ETA, endotracheal aspirate; VAP, ventilator associated pneumonia.

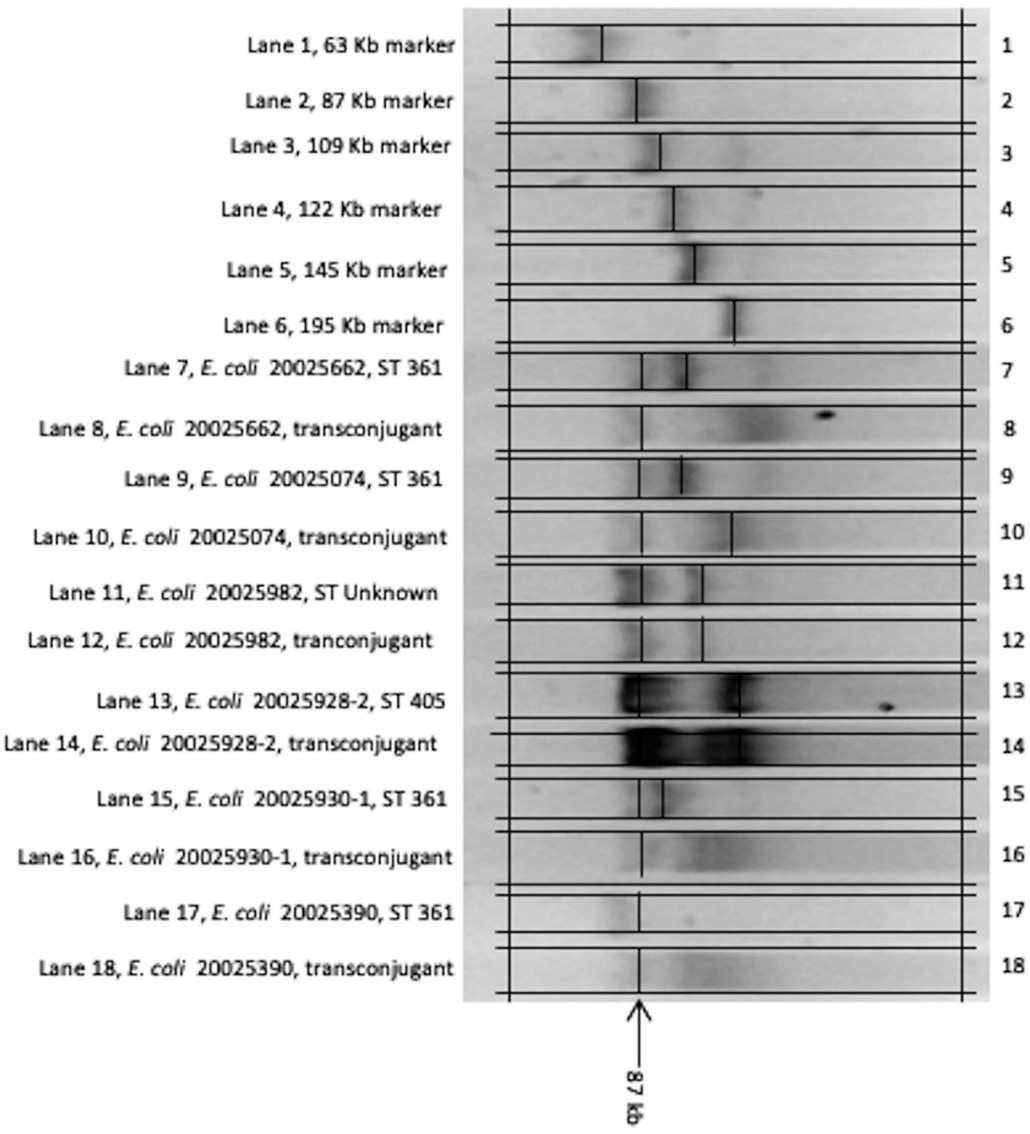

Lane 1, 63 Kb marker — 1
Lane 2, 87 Kb marker — 2
Lane 3, 109 Kb marker — 3
Lane 4, 122 Kb marker — 4
Lane 5, 145 Kb marker — 5
Lane 6, 195 Kb marker — 6
Lane 7, *E. coli* 20025662, ST 361 — 7
Lane 8, *E. coli* 20025662, transconjugant — 8
Lane 9, *E. coli* 20025074, ST 361 — 9
Lane 10, *E. coli* 20025074, transconjugant — 10
Lane 11, *E. coli* 20025982, ST Unknown — 11
Lane 12, *E. coli* 20025982, tranconjugant — 12
Lane 13, *E. coli* 20025928-2, ST 405 — 13
Lane 14, *E. coli* 20025928-2, transconjugant — 14
Lane 15, *E. coli* 20025930-1, ST 361 — 15
Lane 16, *E. coli* 20025930-1, transconjugant — 16
Lane 17, *E. coli* 20025390, ST 361 — 17
Lane 18, *E. coli* 20025390, transconjugant — 18

87 kb

**FIG 2** Conjugation of $bla_{NDM-1}$-carrying 87-kb plasmid from patient isolates to *E. coli* strain J53. The molecular-weight markers were obtained from *Rhizobium* spp. genomic DNA.

We describe an outbreak of NDM-1-producing *E. coli* among critically ill COVID-19 patients. The multilocus sequence typing and PFGE results are highly suggestive of clonality among the isolates recovered from affected patients. There was a high prevalence of CRE colonization in the affected ICU.

$bla_{NDM-1}$ was detected in *E. coli* isolates of three different sequence types. The isolates of all three sequence types harbored an IncFII plasmid whose transmission through conjugation resulted in transmission of the $bla_{NDM-1}$ gene. This confirms that the $bla_{NDM-1}$ gene was contained in the IncFII plasmid and is highly suggestive of horizontal transmission of $bla_{NDM-1}$ among different *E. coli* strains. IncFII plasmids are the most abundant plasmid type in members of the *Enterobacterales*. They are responsible for widespread dissemination of $bla_{CTX-M-15}$ and may harbor $bla_{NDM-1}$. Plasmid-mediated resistance is concerning because it can spread through horizontal transmission between organisms and even between different bacterial genera (3, 4, 9, 10).

The mortality rates of CRE infections have been reported to range from 30 to 80% (2, 11). The few active antimicrobials are characterized by their toxicity (colistin) or suboptimal efficacy (tigecycline) (12, 13).

Multiple factors contributed to this outbreak. The demands placed by the surge of patients altered the usual hospital workflow. Surveillance of infection prevention practices and the activities of the antimicrobial stewardship (AMS) program were disrupted. New staff were employed in the repurposed ICUs because senior staff with comorbidities went on temporary paid leave. Cumbersome PPE (double-gloving and gowning) due to fear of contagion hindered hand washing. Lack of information on bacterial coinfection in COVID-19 patients resulted in widespread inappropriate empirical antimicrobial use due to concerns about bacterial coinfection. Finally, guidelines for VAP prevention do not address patients ventilated in the prone position (14). These factors combined to create a permissive setting for the selection, amplification, and dissemination of a clonal strain of CR *E. coli*. The isolate's origin is unknown.

Following the outbreak, bed capacity in the affected unit was reduced from 20 to 14 and infection control measures were reinforced. Feedback on the proper use of PPE was provided, and strict AMS reintroduced. No CRE infections were detected after the interventions.

Similar outbreaks involving *Candida auris* and *Acinetobacter baumannii* have been reported in other centers caring for COVID-19 patients (15, 16) Double-gloving was thought to have contributed to the *C. auris* outbreak, while excessive workload and PPE shortage drove the *A. baumannii* one. Pandemic response policies should emphasize the need to preserve infection control practices to limit collateral damage from disruptions in the usual hospital workflow. Research on how to adapt prevention bundles to COVID-19 patients is necessary.

The limitations of these study were the failure to recover all isolates for molecular testing and the cross-sectional intervention. Patients admitted after this time frame were not screened, so we cannot determine how long CRE colonization lasted.

In conclusion, we documented an outbreak of clonal NDM-1-producing *E. coli* and dissemination of carbapenem resistance through a mobile element during hospital conversion for the COVID-19 pandemic. Reinforcement of infection control practices helped contain the outbreak.

## MATERIALS AND METHODS

**Isolate investigation and screening.** Isolates were identified using the BD Bruker matrix-assisted laser desorption ionization (MALDI) Biotyper (Bruker Daltonic, Inc., Billerica, MA, USA). Antimicrobial susceptibility testing (AST) was done with the Vitek-2 instrument (bioMérieux, Marcy l'Etoile, France). Carbapenem and colistin MICs were determined with broth microdilution following CLSI recommendations (7).

Carbapenemase detection was performed using the modified carbapenem inactivation method (mCIM). Metallo-$\beta$-lactamase was detected using the EDTA-modified carbapenem inactivation method (eCIM) (6, 7).

Enzyme genes were identified with real-time PCR using the CRE ELITe MGB kit (Elitech Group, Svizzera, Turin, Italy) following the manufacturer's instructions. The kit can detect genes of the KPC, NDM, VIM, IMP, and OXA-48-like families.

Rectal swabs were taken from ICU patients to screen for CRE colonization. Swabs were deposited in Trypticase soy broth (BBL; Becton, Dickinson, Sparks, MD, USA) supplemented with a 10-$\mu$g ertapenem disk (Becton, Dickinson) and incubated at 35 $\pm$ 2°C for 24 h. The bacteria that grew were identified, their antimicrobial susceptibilities were determined, and they were tested for carbapenemase carriage (8).

**Clonality assessment.** Carbapenemase-producing *E. coli* isolates were assessed for clonality using MLST. The following genes were amplified: *adk*, *fumC*, *gyrB*, *icd*, *mdh*, *purA*, and *rec*. Amplicons were purified by using QIAquick PCR purification spin columns (Qiagen, Venlo, Netherlands) and sequenced with a Genetic Analyzer 3500 automated sequencer (AB Applied Biosystems, Hitachi, San Francisco, CA, USA). Sequences were aligned using BioEdit software (Ibis Bioscience, Carlsbad, CA, USA) and compared with the MLST database (MLST locus sequence definitions are available online at https://pubmlst.org/bigsdb?db=pubmlst_ecoli_achtman_seqdef). Allele numbers were determined, and the sequence type (ST) was assigned.

Genetic relatedness was determined using PFGE. The genomic DNA of the *bla*~NDM-1~-positive *E. coli* isolates and reference marker *Salmonella enterica* subsp. *enterica* serovar Braenderup strain ATCC BAA-664 were digested by XbaI endonuclease and run through 1% agarose gels in a CHEF Mapper XA PFGE system (Bio-Rad, USA). The gels were stained with ethidium bromide and visualized using the GEL logic 1500 imaging system (Kodak, Rochester, NY, USA). The PFGE profiles were analyzed with BioNumerics software version 7.6 (Applied Maths, Sint-Martens-Latern, Belgium).

**Plasmid analysis.** PCR-based replicon typing (PBRT) was performed on isolates carrying *bla*~NDM-1~ (3, 9, 10, 17). DNA was extracted using the Qiagen DNA minikit (Qiagen, GmbH, Hilden, GA, USA). PCR was

performed using a VeritiPro thermal cycler (Applied Biosystems, Foster City, CA, USA) following the protocol described by Villa et al. (9) Amplicons were run through 1% agarose gels with SYBR green and read using the GEL logic 1500 imaging system (Kodak).

For conjugation experiments, two isolates from respiratory samples and 3 from rectal samples were selected. The isolates corresponded to ST361, ST405, and ST unknown. Ertapenem resistance was carried out using the mating method on nitrocellulose filters (18). Sodium azide-resistant *E. coli* strain J53 was used as the recipient. Donor and recipient strains were grown separately in 5 mL of Luria-Bertani broth (Hardy Diagnostics, Santa Maria, CA, USA) overnight at 37°C. The donor and recipient cultures were mixed at a 1:2 ratio in a nitrocellulose filter over a Luria-Bertani agar plate and incubated at 37°C for 8 h. Transconjugants were selected on Luria-Bertani agar containing 100 mg/L of sodium azide and 1 mg/L of ertapenem and ceftazidime (Sigma, St. Louis, MO, USA). $bla_{NDM-1}$ carriage was confirmed on *E. coli* transconjugants using endpoint PCR.

## ACKNOWLEDGMENTS

We acknowledge Martha Asunción Huertas-Jiménez, Roxana de Paz-García, Anabel Haro-Osnaya, and Alma Rosa Chávez-Ríos for their invaluable work in infection control. They have gone to great lengths in keeping both patients and health care personnel safe during these trying times.

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
