## [Reviewer comments · Microbiology Spectrum]

Microbiology Spectrum

Outbreak of NDM-1 producing *Escherichia coli* in a COVID-19 intensive care unit in a Mexican tertiary-care center.

Oscar Fernández-García, Maria F. González-Lara, Marco Villanueva-Reza, Nereyda de-León-Cividanes, Luis F. Xancal-Salvador, Verónica Esteban-Kenel, Ayleen Cárdenas-Ochoa, Axel Cervantes-Sánchez, Areli Martínez-Gamboa, Eric Ochoa-Hein, Arturo Galindo-Fraga, Miriam Bobadilla-del Valle, Jose Sifuentes-Osornio, and Alfredo Ponce-de-León

Corresponding Author(s): Alfredo Ponce-de-León, Clinical Microbiology Laboratory/Department of Infectious Diseases of Salvador Zubirán National Institute of Medical Sciences and Nutrition, Mexico City, Mexico

Review Timeline:

Submission Date:

December 3, 2021

Accepted:

December 15, 2021

Editor: Daria Van Tyne

Reviewer(s): The reviewers have opted to remain anonymous.

Transaction Report:

DOI: <https://doi.org/10.1128/spectrum.02015-21>

December 15, 2021

Dr. Alfredo Ponce-de-León
Clinical Microbiology Laboratory/Department of Infectious Diseases of Salvador Zubirán National Institute of Medical Sciences and Nutrition, Mexico City, Mexico
City of Mexico
Mexico

Re: Spectrum02015-21 (Outbreak of NDM-1 producing *Escherichia coli* in a COVID-19 intensive care unit in a Mexican tertiary-care center.)

Dear Dr. Alfredo Ponce-de-León:

Your manuscript has been accepted, and I am forwarding it to the ASM Journals Department for publication. You will be notified when your proofs are ready to be viewed.

Thank you for addressing the reviewer comments from your prior review at AAC. I have one additional request, which is that you include source (patient and anatomic source) of each isolate in the PFGE analysis shown in Fig. 1.

Sincerely,

Daria Van Tyne
Editor, Microbiology Spectrum
